# Brain Long Noncoding RNAs: Multitask Regulators of Neuronal Differentiation and Function

**DOI:** 10.3390/molecules26133951

**Published:** 2021-06-28

**Authors:** Sarva Keihani, Verena Kluever, Eugenio F. Fornasiero

**Affiliations:** Department of Neuro- and Sensory Physiology, University Medical Center Göttingen, 37073 Göttingen, Germany; sarvakeihani@gmail.com (S.K.); verena.klver@stud.uni-goettingen.de (V.K.)

**Keywords:** long noncoding RNAs, neurons, neuronal development, neuronal differentiation, neurogenesis, synaptic activity, synaptic plasticity

## Abstract

The extraordinary cellular diversity and the complex connections established within different cells types render the nervous system of vertebrates one of the most sophisticated tissues found in living organisms. Such complexity is ensured by numerous regulatory mechanisms that provide tight spatiotemporal control, robustness and reliability. While the unusual abundance of long noncoding RNAs (lncRNAs) in nervous tissues was traditionally puzzling, it is becoming clear that these molecules have genuine regulatory functions in the brain and they are essential for neuronal physiology. The canonical view of RNA as predominantly a ‘coding molecule’ has been largely surpassed, together with the conception that lncRNAs only represent ‘waste material’ produced by cells as a side effect of pervasive transcription. Here we review a growing body of evidence showing that lncRNAs play key roles in several regulatory mechanisms of neurons and other brain cells. In particular, neuronal lncRNAs are crucial for orchestrating neurogenesis, for tuning neuronal differentiation and for the exact calibration of neuronal excitability. Moreover, their diversity and the association to neurodegenerative diseases render them particularly interesting as putative biomarkers for brain disease. Overall, we foresee that in the future a more systematic scrutiny of lncRNA functions will be instrumental for an exhaustive understanding of neuronal pathophysiology.

## 1. Introduction

Despite being initially considered as ‘transcriptional noise’, lncRNAs are currently recognized as modulators of cellular functions in every branch of life and have been implicated in a large number of biological processes including X-chromosome silencing, genetic imprinting, regulation of cell state, coordination of differentiation and possibly disease modulation (reviewed in [1,2,3,4,5,6]). This concept is supported by the carefully-regulated subcellular localization of lncRNAs, with several examples of lncRNAs that are restricted to the nucleus [7], while others are transported into the cytoplasm [8,9]. Likewise, the number of lncRNAs in different organisms seems to correlate with genome complexity [1], and in complex tissues such as the nervous system, lncRNAs show extreme spatiotemporal regulation and cell specificity [8,10,11,12]. In this review we focus on the incipient roles of lncRNAs as flexible regulators of neuronal activity and brain pathophysiology.

## 2. General Characteristics of LncRNAs

While only a small portion (~3%) of the vertebrate genome encodes for proteins, it has been proposed that ~80% can lead to the production of a variety of noncoding RNAs (ncRNAs) [13,14,15,16,17]. It is still debated whether the pervasive transcription leading to the production of these molecules is noise related to the regulation of the expression of neighboring genes, a way to provide ‘raw-material’ for the evolution of de novo genes or rather leads to the production of functional molecular species [18,19,20,21,22,23,24,25]. The large family of ncRNA comprises several RNA species including (i) the classical ncRNAs involved in translation and pre-mRNA processing (such as tRNAs, rRNAs and small nuclear RNAs); (ii) molecules of 21–28 nucleotides that regulate transcription and mRNA silencing (small interfering RNAs and miRNAs) and (iii) a more heterogenous family of longer transcripts (>200 nucleotides) collectively defined as long noncoding RNAs (lncRNAs) [26,27,28]. Several lncRNAs share similarities with conventional mRNAs, such as being generally synthetized by RNA polymerase II, undergoing capping, splicing and polyadenylation. However, lncRNAs are usually less abundant than most of conventional mRNAs and by definition display virtually non-existent coding potential [6,29]. The absence of coding potential needs to be considered carefully when studying newly-discovered lncRNAs, as it is becoming clear that appeared initially devoid of functional open reading frames are actually involved in the production of small peptides, revealing a hidden ‘short proteome‘ which has important cellular functions, such as transcription regulation or cell signaling [30,31,32,33,34]. In general, a real challenge in this field is to distinguish transcriptional noise from molecules that exert a ‘true molecular function’. In this review, whenever possible, we have tried to follow experimental evidences that would help us to distinguish these two scenarios (noise vs. function). While for some few extensively studied lncRNAs this is possible, the task is complicated by the fact that often there are few follow up works on the same lncRNA and the literature is growing exponentially with newly found lncRNAs. Here we have summarized the more relevant lncRNAs in brain function.

## 3. LncRNAs in the Nervous System

The brain of vertebrates, and in particular of birds and mammals, is one of the most complex systems in biology where an elaborate network established between different cell types allows advanced cognitive tasks [35]. During brain development, an intricate series of processes is required for cell differentiation and construction of refined neuronal circuits. Even in the adult brain, processes such as memory recall, learning and the underlying orchestration of cell activities require an exceptionally precise spatiotemporal tuning of gene expression and protein production. Thanks to their structural plasticity and intrinsic ability to facilitate protein-genome interactions, it is becoming evident that lncRNAs might be optimally suited for the regulation of brain gene expression, as we outline in this review.

The importance of lncRNAs in the central nervous system is exemplified by the high number of brain-specific transcripts compared to other tissues [36,37], i.e., up to 40% of the differentially expressed lncRNAs are brain-specific [38]. Several transcriptome studies identified thousands of lncRNAs expressed during brain development [39,40,41,42,43]. Measures of RNA abundance also revealed changes of lncRNA expression during neurogenesis [40,42,44], and specific expression patterns in distinct neuronal sub-types suggest that they could be involved in cell fate choices [37,45]. Numerous lncRNAs that vary during development are embedded in the genome in the vicinity of neurodevelopmental regulators [46]. While this observation could also be explained by leaky transcription in the vicinity of active genomic regions, several examples suggest that lncRNAs tend to be co-regulated with other genes, as often happens with ncRNAs whose expression is coordinated in gene clusters [47,48].

Results from in situ hybridization and *lacZ* reporter mouse lines have revealed that lncRNAs are excellent markers for brain developmental stages, brain sub-region identity and even reliable markers for neuronal subtypes, further indicating that there is a tight regulation in their expression profile and that they are intentional transcription products [8,11,12,49]. Moreover, besides being modulated during differentiation, hundreds of lncRNAs are dynamically controlled by neuronal activity and are associated with the dynamic regulation of synapses [50,51].

Despite the overall low level of conservation of lncRNAs, brain-specific lncRNAs display high sequence conservation when compared to lncRNAs from other tissues [46,52]. Even their spatiotemporal expression is maintained among some species as confirmed by a set of human lncRNAs showing transient expression during primate organoid differentiation, as well as cell-specific expression shown by single-cell sequencing [53]. Even more astoundingly, some neurospecific lncRNAs are conserved across species with larger brains, suggesting a specific role in the evolution of the primate neocortex [54]. Consistently, there are significant structural similarities between the ~3000 multi-exonic lncRNAs found in humans with those of great apes and also in their expression profile in cortical development [53].

Altogether, these observations remain correlative, raising the question of whether lncRNA expression is just an ‘epiphenomenon’, a casual correlate of other processes such as gene expression regulation, or if it plays a direct causative role in the regulation of brain morphogenesis and functioning. To answer this question, in the next sections we will summarize the molecular roles of the most relevant lncRNAs and their functions in neurons and other cells for the central nervous system.

## 4. Functional Classification of Neuronal LncRNAs

Among the numerous possible classifications for lncRNAs proposed [55,56,57,58,59,60,61], here we decided to divide brain lncRNAs in six main functional classes. These include: (1) *Enhancer lncRNAs* (eRNAs), which are the product of bidirectional DNA transcription taking place in the region of active enhancers [62]; (2) *Antisense lncRNAs* (AS-lncRNAs; also known as natural antisense transcripts) produced by antisense transcription of a protein-coding sequence which can be degraded following the formation of double strand RNAs [63]; (3) *Cis-acting nuclear lncRNAs*, thus acting in the same genome locus from which they originate and that interfere with and/or regulate the transcription of neighboring genes [64]; (4) *Trans-acting nuclear lncRNAs* involved in epigenetic regulation; (5) *LncRNAs that modulate gene expression by post transcriptional mechanisms* and affect mRNA stability, translation, or splicing [65,66,67] and finally (6) *LncRNAs directly modulating protein functions* that are generally functioning in association with RNA binding proteins (RBPs), in some cases also outside the nucleus. This classification is summarized in Figure 1.

### 4.1. Enhancer LncRNAs: More Than Markers of Actively Transcribed Regions

Even though eRNAs are sometimes excluded from the class of lncRNAs because they can be shorter than 200 nucleotides, the vast majority are longer than 200 nucleotides. Moreover, these ncRNAs do not behave as most other shorter ncRNAs. For these reasons, and for their predominant abundance in the brain, we believe they should be discussed here.

Several eRNAs are produced in the vicinity of genes as bidirectional transcripts, whose levels correlate with the local gene expression [68,69]. For this reason eRNAs were originally considered markers of actively transcribed loci [68]. In 2010, Greenberg and collaborators showed that in neurons ~12,000 enhancers are actively transcribed and give rise to eRNAs upon neuronal activation in a process requiring polymerase II, the p300-CBP coactivator, and triggering the mono-methylation of Histone 3 at lysine 4 [68]. In neurons, the synthesis of eRNAs facilitates the recruitment of activity-regulated transcription factors including CREB, SRF, NPAS4 and leads to the formation of enhancer-promoter loops, driving the transcription of activity-regulated genes such as *c-Fos* and *Nr4a2* [68,70,71].

For the activity-regulated cytoskeleton-associated gene (*Arc*), it was described that eRNAs can directly modulate in *cis* the elongation of the corresponding mRNA by acting as decoys for the negative elongation factor NELF, thus facilitating the progression of RNA polymerase II at the promoter of the target gene [72]. Similar mechanisms could underlie the function of *utNgn1*. eRNA *utNgn1* is expressed in concomitance with Neurogenin-1 (NGN1), an essential transcription factor for neuronal differentiation [73], and loss of function experiments indicate that *utNgn1* might control the levels of NGN1 [73].

Altogether these findings point to a specific biological function of eRNAs, suggesting that their transcription is not just a side effect of the change in the local DNA conformation and epigenetic state, but that they might directly recruit protein complexes. They might even function as bridges between far DNA regions. This is exemplified by *Evf2*, a highly-conserved eRNA with a role in brain development. Loss of *Evf2* in mice is accompanied by a reduction in GABAergic interneurons in the newborn hippocampus followed by impaired GABA circuitry in adult animals. In the developing forebrain, *Evf2* has a complex regulatory mechanism since it can simultaneously localize and activate a gene distant ~1.5 Mb on chr6 (*Umad1*), while repressing another gene that is ~25 Mb distant (*Akr1b8*). Moreover, its overexpression activates *Lsm8*, also on the same chromosome but ~12 Mb apart [74]. These effects are probably mediated through complex changes of three-dimensional chromosome topology, although other effects in *trans* cannot be excluded, since it has been observed that *Evf2* can also recruit transcriptional activators (such as DLX1/2) and repressors (such as MECP2) that can interfere with the regulatory regions of developmental genes such as *Dlx5*, *Dlx6* and *Gad1*, explaining its effect on GABAergic differentiation [75].

Overall, a comprehensive study of the molecular functions and structural effects of eRNAs is largely missing. Remarkably, several single nucleotide polymorphisms on eRNAs near synaptic activity-regulated genes are associated with autism spectrum disorders, indicating that their sequences contain structural motifs required for a specific scaffolding function [76].

### 4.2. Antisense LncRNAs Regulate Key Genes in the Brain

Around 70% of genes are transcribed in the antisense direction, giving rise to AS-lncRNAs implicated in post-transcriptional regulation of the corresponding ‘sense’ transcript by degradation of the complementary mRNA molecules or by modulating their expression or stability [77,78,79,80,81].

In the brain, several mRNAs are under the control of a corresponding AS-lncRNA, including for example the genes encoding the CAMK2N1 subunit of the calmodulin-dependent protein kinase and neurogranin (NRGN), which are both implicated in the regulation of postsynaptic signal transduction pathways and synaptic long-term potentiation [82].

Another notable example is *Kcna2-AS* that targets the mRNA of the ‘shaker type’ voltage-dependent potassium channel (*Kcna2*), which is relevant for human pathology since its mutations are linked to ataxia and focal epilepsies [83]. Indeed, the overexpression of *Kcna2-AS* in experimental models inversely correlates with the levels of KCNA2 [84], confirming that this AS-lncRNA can regulate its respective protein. In rats, peripheral nerve injury increases the expression of *Kcna2-AS* as a result of transcription activation mediated by the myeloid zinc finger protein 1 (MZF1) that leads to a ~50% decrease of active channels [84]. As a consequence, the induction of *Kcna2-AS* during nerve damage reduces the voltage-gated potassium current, and increases the excitability of dorsal root ganglia neurons producing neuropathic pain symptoms. While these observations hold great potential for therapeutic strategies aimed at decreasing neuropathic pain following injury, this does not entirely explain the physiological roles of *Kcna2-AS* in the regulation of neuronal excitability which will require further studies.

During the development of the brain cortex, several AS-lncRNA are expressed to subtly regulate the amounts of transcription factors. Examples include *Sox* (SRY-related HMG-box) genes that encode high mobility groups that can bind and bend the DNA during the regulation of embryonic development and influence cell fate. In particular two members of the *Sox* family, *Sox4* and *Sox11*, have corresponding AS-lncRNAs that are highly expressed during the development of the mouse cortex, suggesting they might have a specific function in cerebral corticogenesis and cell differentiation [85].

An eminent AS-lncRNAs in the brain is *BDNF-AS*, which controls the levels of the brain-derived neurotrophic factor (BDNF), a neurotrophin implicated in the regulation of several key processes including neurogenesis, precursor cell proliferation, neuronal differentiation, maturation, plasticity and with a role in pathophysiology of psychiatric disorders [86,87,88,89]. *BDNF-AS* overexpression decreases the levels of BDNF, both in vitro and in vivo [63]. The exact mechanisms of *BDNF-AS* function are not entirely known, although it seems that its expression affects in *cis* the *Bdnf* gene by recruiting EZH2, an abundant histone methyltransferase that causes the inactivation of the BDNF promoter. As expected, *BDNF-AS* knockdown results in increased neuronal cell number, differentiation and more prominent neurite outgrowth [63,90]. The postmortem amygdala of alcoholics showed that the expression of *BDNF-AS* is increased only in individuals who started to drink before 21 years of age, suggesting that this *BDNF-AS* might affect epigenetic reprogramming in an age-dependent manner [91]. Deficits of BDNF signaling in the amygdala decreased the levels of the activity-regulated cytoskeleton-associated protein (ARC) and reduced synaptic plasticity while increasing the risk of alcohol dependence in adulthood [91].

Among AS-lncRNAs associated with human pathologies, it is worth mentioning *BACE1-AS* which, in contrast to most AS-lncRNAs, stabilizes its related transcript *Bace1* which encodes the β-secretase enzyme 1 implicated in Alzheimer’s disease (BACE1) [80]. This enzyme cuts and processes the amyloid precursor protein (APP), and the *BACE1-AS* lncRNA is thus involved in a feed-forward loop that is induced by high levels of the amyloid-beta 1–42 fragment itself. The molecular details of this unusual process are not known and, although it has been suggested that RNA duplex formation could alter the structure of *Bace1* RNA and thus increase its stability, this possibility remains speculative. To reinforce its connection to the pathology, the concentration of *BACE1-AS* is higher in patients with Alzheimer’s disease than in matched controls [80] and correlates with Aβ production and plaque deposition in animal models [92].

Angelman syndrome is a pathology where an AS-lncRNA, named *Ube3a-ATS*, might become a possible target for therapeutic intervention. This transcript is nuclear, neurospecific and regulates the gene *Ube3a* [93]. Angelman syndrome is a neurodevelopment disorder where patients develop intellectual disability, speaking and movement problems often associated with what looks like a happy demeanor [94]. The gene *Ube3a* codes for a ubiquitin ligase, that ubiquitinates and regulates the stability of several important proteins in neuronal function such as MAPK1 and β-catenin. The paternal *Ube3a* is usually repressed by *Ube3a-ATS* and the pathology manifests itself upon deletion of the maternal gene. As a consequence of paternal imprinting, cells that in principle contain a copy of the functioning gene, do not actually express the protein UBE3A. For this reason, downregulation of the paternal *Ube3a-ATS* could be a rescue strategy for this pathology [95,96].

A number of AS-lncRNAs appear to have acquired additional functions besides regulating the expression of their sense gene. One example is *Six3OS*, which is transcribed from the opposite strand of the *Six3* gene, a homeodomain transcription factor involved in forebrain development and retinal cell specification [97]. Besides regulating the expression of *Six3*, *Six3OS* also acts as a molecular scaffold for the transcriptional co-regulator of SIX3 (EYA) and the histone-lysine N-methyltransferase enzyme EZH2, both implicated in the regulation of SIX3 target genes [98]. Interaction of *Six3OS* with EZH2 possibly changes the chromatin structure by triggering the H3K27me3 modification, which is usually associated with decreased expression of nearby genes [98]. Furthermore, loss of function studies revealed that *Six3OS* is implicated in the glial-neuronal lineage specification of adult neural stem cells in vivo [41]. Due to technical challenges related to the study of antisense RNA, it is conceivable that there is some skepticism for results that could also be simply due to an effect on the target gene (i.e., *Six3*). An additional confounding factor could be the several isoforms of *Six3OS*, some of which probably code for short ~100 amino acid proteins [99]. Nevertheless, it can be hypothesized that lncRNAs that were initially expressed for the direct regulation of their respective mRNA with an antisense mechanism acquired during evolution additional specific functions in *cis* in the vicinity of the gene, similar to *cis*-acting lncRNAs which will be discussed in the next section.

### 4.3. Cis-Acting LncRNAs Regulating Nearby Genes Might Evolve Additional Molecular Functions

The local regulation of gene expression exerted by lncRNAs can encompass different mechanisms, such as regulation of promoter activity but also local modulation of transcription and splicing [100,101]. The reason why lncRNAs might restrict their function at the site of their production is not entirely clear, although it might be related to the fact that once they are produced, some lncRNAs are physically engaged in local complexes, thus limiting their range of action. This might be the case for several lncRNAs that are operating locally at the chromatin interface [102,103]. Since it is known that several lncRNAs interfere with transcription and translation in the vicinity of their genes, this is the first mechanism that is investigated for lncRNAs and their regulatory role in *cis* has been observed and described in many cases. At the same time, in several cases as also summarized here, a more careful examination of their role has revealed that many lncRNAs also have additional roles, besides their activity in *cis*.

A relevant lncRNA with a *cis*-acting mechanism is *Paupar* (*Pax6* upstream antisense RNA), which is transcribed upstream the *Pax6* gene in the opposite direction [104]. *Paupar* acts in *cis* regulating PAX6-dependent functions since its knockdown leads to an increase in PAX6 expression and interferes with eye development, progenitor cell potency and proliferation in the brain [104,105]. The exact molecular mechanism of regulation on the *Pax6* is not entirely known, although it could be speculated that its transcription affects the nearby promoter region of *Pax6*, either stabilizing transcription factor binding or inducing promoter-enhancer looping. In parallel, *Paupar* has probably evolved a more elaborate mechanism of action. In addition to regulating the expression of *Pax6*, it also regulates the binding of PAX6 to its neural target genes [105] and acts in *trans* by forming a ribonucleoprotein (RNP) complex together with PAX6 and KAP1 (coded by *Trim28*), which is involved in transcriptional control [104,106]. It seems that *Paupar* promotes the association of KAP1 with the DNA resulting in H3K9me3 modifications on the promoter of genes involved in neural stem cell renewal and differentiation, inducing heterochromatin and inactivation [106]. Loss of function studies confirm that the lack of *Paupar* disrupts olfactory bulb neurogenesis [106] and induces neuroblastoma cell differentiation [104].

A key transcription factor in cerebral cortex development is POU3F3 (also known as BRN1). It has been reported that POU3F3 is under the control of several lncRNAs. In its close proximity, there are two lncRNAs that have been studied in detail by the laboratory of Rinn and collaborators [11]. Deletion of one of these two lncRNAs, *linc–Brn1b* (also known as *Pantr2*), has a *cis*-regulatory effect on *Pou3f3*. The level of *linc–Brn1b* usually is higher in neural stem cells and its expression declines during development. Mice where both copies of *linc–Brn1b* are ablated exhibit a ~50% reduction in POU3F3 levels and a subsequent decrease in the number of cortex intermediate progenitor cells and abnormal lamination in the cerebral cortex [11]. While the mechanism of action of this lncRNA is not entirely known, its close proximity to the *Pou3f3* gene and the effect on its expression suggest that the *linc–Brn1b* acts as an eRNA, although the differential expression observed in other surrounding genes might suggest that *linc–Brn1b* also has additional functions that surpass its local effects on DNA conformation. *Dali* (DNMT1-Associated Long Intergenic RNA) is yet another lncRNA that interferes with the expression of *Pou3f3*, which is located ~50 Kb upstream. Additionally, similarly to *Paupar*, *Dali* has a set of supplementary functions and forms a complex with POU3F3 which, in *trans,* regulates a subset of genes involved in neural differentiation. Another mechanism by which *Dali* carries on its functions is achieved by its direct interaction with chromatin remodeling factors such as BRG1, SIN3A and DNMT1 [64]. Through these interactions *Dali* affects the epigenetic reprogramming of gene promoters, as also shown by its knockdown which increases the methylation of CpG islands on the promoters of genes involved in neuronal differentiation [64]. There are also several other lncRNAs which have a role in the epigenetic modulation of the brain that will be discussed more in detail in the next section.

### 4.4. Trans-Acting LncRNAs Involved in Epigenetic and Transcriptional Regulation

In several tissues, nuclear lncRNAs often act as molecular scaffolds that recruit chromatin modifiers such as the polycomb repressive complex (PRC2) and the DNA methyltransferase 3 (DNMT3) to regulatory genomic elements and modulate the expression of target genes (Reviewed in Refs. [107,108]). In the brain, global dynamic modification of chromatin is often observed during cell lineage commitment, development, terminal differentiation and in more complex cognitive functions such as learning [109,110,111]. While lncRNAs do not have overt ribozyme-like ‘enzymatic’ functions, their interaction with the epigenetic machinery and the target DNA sequences coupled to their ability to act as a scaffold for the formation of large multimeric complexes, can lead to precise covalent modification of histones and/or DNA and result in the control of gene expression. As an example, *lncOL1* is a lncRNA with epigenetic functions, which is exclusively expressed in oligodendrocytes, a specific type of glial cells deputed to myelination in the central nervous system. *lncOL1* promotes oligodendrocyte differentiation and speeds the onset of myelination forming a complex with SUZ12, a zinc finger that is part of the polycomb repressive complex 2. The *lncOL1*-SUZ12 complex silences genes that in undifferentiated cells repress oligodendrocyte genes promoting the formation of H3K27me3 modifications on their regulatory elements. As a result, *lncOL1* overexpression promotes myelination and even more strikingly favors re-myelination following injury, suggesting a possible therapeutic use of this lncRNA in the treatment of demyelinating conditions [112].

Regulating neuronal differentiation and neurogenesis, examples of lncRNAs include the Hox-encoded *HOXA Transcript Antisense RNA, Myeloid-Specific 1* (*HOTAIRM1*) and *Tcl1* upstream neuron-associated lincRNA (*Tuna*). *HOTAIRM1* modulates Neurogenin 2, a basic helix-loop-helix transcription factor that acts as a master regulator of neurogenesis [113]. Downregulation of *Tuna*, a highly expressed and evolutionary conserved lncRNA that has been studied in the developing mouse brain, is sufficient to block neurogenesis in both mouse and human cell models [114].

Tuna affects neuronal development by recruiting two heterogenous nuclear ribonucleoproteins (HNRNPK and PTBP1) and Nucleolin, which can regulate chromatin condensation by bringing histone H1 in the proximity of genes to be silenced [114]. The overexpression of *Tuna* results in the deposition of the active histone modification H3K4me3 to the promoters of key genes in neuronal specification such as *Nanog*, *Fgf4* and *Sox2*, indicating that the lncRNA might also modulate the activity of some histone methyltransferases [114].

Transcriptional regulation can take place independently from epigenetic remodeling as possibly observed in the case of the lncRNA *GM12371*. This lncRNA is nuclear and more abundantly expressed in the hippocampus. Its depletion in primary hippocampal neurons determines the reduction of spontaneous excitatory postsynaptic currents. Furthermore, decreasing *GM12371* interferes with spine density and dendritic arborization. Mechanistically, *GM12371* is regulated by cAMP-PKA signaling and, once produced, it binds to active chromatin in *trans* to regulate genes which are involved in neuronal growth and development. The expression of *GM12371* is strictly required in neurons to induce changes in excitatory synaptic transmission mediated by cAMP-PKA signaling [115].

Another example of a lncRNA involved in transcriptional regulation is *RMST,* which was originally discovered as a rhabdomyosarcoma associated transcript. During neuronal differentiation, *RMST* is significantly upregulated in human embryonic stem cells and it is able to modulate the expression of neurogenic transcription factors [116,117]. *RMST* is under the control of the RE1-silencing transcription factor (REST), which is a master repressor of thousands of neurospecific genes [118]. During neuronal differentiation, the repression of REST on *RMST* expression is released and *RMST* interacts with SOX2, the transcription factor that also interacts with *Tuna* and that has an important role in neural fate determination [114]. The complex *RMST*-SOX2 regulates a large number of downstream genes which are involved in neurogenesis, and *RMST* is particularly required for the binding activity of SOX2 to the promoter of target genes [117].

### 4.5. LncRNAs Regulating mRNA Stability, Translation, or Splicing

Another mechanism employed by lncRNAs to interfere with cellular functions is to regulate RNA splicing. The lncRNA *Malat1* is expressed in various tissues and in the brain its levels are particularly high in the latest stages of development, when *Malat1* is found within nuclear speckles [119]. *Malat1* is able to recruit pre-mRNA-splicing factors to active transcription sites [120]. Gene expression profiling of *Malat1*-depleted HeLa cells shows that this lncRNA can interfere with the splicing of pre-mRNAs including the RNA of the calcium/calmodulin dependent protein kinase II beta (*Camk2b*), which is crucial for postsynaptic signaling, plasticity and synaptogenesis [67]. DNA microarray analysis in *Malat1*-depleted neuronal cells also showed that this lncRNA affects the expression of presynaptic proteins (such as SNAP25 and VAMP2) and other proteins relevant for neuronal activity and development, including the activity-regulated cytoskeleton-associated protein (ARC) and the axon guidance co-receptor Neuropilin-1. Additionally, *Malat1* regulates synapse formation by altering transcript levels of the synaptic cell adhesion molecule-1 (SynCAM1) and the postsynaptic cell surface protein Neuroligin-1 [120] and neuritogenesis interfering with the mir-30/Spastin axis [121]. The observation that *Malat1* is important for the regulation of mRNAs encoding synaptic proteins is further confirmed by the fact that synaptic density is reduced by *Malat1* knock-down, while, when *Malat1* is overexpressed, the density of presynaptic boutons increases [120]. Puzzlingly, *Malat1* knockout mice show no distinct phenotype or histological abnormalities [122,123,124] and its depletion does not affect pre-mRNA splicing or global gene expression pattern. To further complicate this picture, a *cis*-regulatory effect of *Malat1* was observed on the close locus of the lncRNA NEAT1 [124]. The exact molecular mechanism of action of *Malat1* remain to be established.

Similar to *Malat1*, the lncRNA *Gomafu* localizes in the nucleus and is expressed in pyramidal neurons in the cerebral cortex and CA1 pyramidal neurons in the hippocampus [125]. Early studies described the role of *Gomafu* in mammalian retinal development and oligodendrocyte lineage specification [125,126]. A significant activity-dependent downregulation of *Gomafu* and subsequent release of splicing factors was observed in transcriptome analyses upon neuronal depolarization in both primary mouse neurons and induced pluripotent stem cell (iPSC)-derived neurons systems [50]. Studies in vivo revealed that *Gomafu* acts as a scaffold for the splicing proteins QKI and SRSF1, and interferes with the splicing of a set of genes involved in embryonic brain development and neural progenitor proliferation including *Wnt7b*, *Disc1* and *Erbb4* [40,50]. In this context *Gomafu* functions to balance proliferation and differentiation of neural progenitor cells through regulating splice variations of genes involved in neuronal cell fate determination [40].

The lncRNA *Pinky* (*Pnky*) is also involved in the early stages of neuronal differentiation. This lncRNA is highly expressed in the ventricular and the subventricular zone of adult mice and it has high levels in the developing brain. *Pnky* regulates neural stem cell self-renewal and its expression is decreased during the differentiation of neural stem cells. Loss of function experiments showed that *Pnky* knockdown increased neurogenesis in both the developing embryonic and postnatal cortex [127]. Functional characterization revealed that *Pnky* interacts with a splicing factor, PTBP1, known as a repressor of neuronal differentiation and its deletion in vivo has no effect on the expression of the nearby *Pou3f2* transcript, advocating for a *bona fide trans* mechanism of activity [128].

LncRNAs can also indirectly increase the expression levels of target mRNAs by acting as molecular sponges or decoys for microRNAs (miRNAs) that, if not blocked, would lead to mRNA degradation. For instance, the neurodevelopment lncRNA ‘*LncND*’ is expressed in neuron progenitor cells and its levels decrease during the differentiation of neurons. *LncND* works as a sponge for *miR-143-3p* and is particularly efficient since it contains 16 complementary elements for this miRNA. As an example, by blocking *miR-143-3p*, *LncND* regulates the levels of the Notch signaling receptors NOTCH-1 and NOTCH-2. Although *LncND* is absent in mice, its overexpression in mouse radial glia cells (RGC) resulted in the expansion of PAX6-positive RGC by interacting with the highly conserved miRNA-143-3p in the developing mouse cortex. These results imply the role of *LncND* in cerebral cortex expansion through miR-mediated regulation of Notch signaling [129]. Another miRNA decoy, implicated in brain development, is *lncRNA-1604*. This lncRNA is necessary for ectoderm differentiation of mouse ESCs. Functional analyses reveal that *lncRNA-1604* is a decoy for miR-200c, and it reduces the degradative potential of miR-200c on the mRNAs of the transcription factor ZEB1/2, which promotes neural differentiation [130]. Another recent example of a lncRNA that functions as a miRNA sponge is *Synage*, which regulates synapse development in the cerebellum by titrating *miR-325-3p* [131]. Through this mechanism, *Synage* increases the expression of the Cerebellin 1 Precursor transcript (*Cbln1*), which is essential for maintaining climbing fiber-Purkinje cell synapses. The KO of *Synage* in mice leads to cerebellar atrophy and neuron loss. *Synage* is one of the few examples of predominantly cytosolic lncRNAs, and it has been suggested to have an additional function as organizer of the synaptic scaffold [131]. Among other examples of lncRNAs acting in a similar manner, it is worth mentioning *Arrl1*, which acts as a sponge of *miR-761* that would decrease the levels of *Cdkn2b* and thus inhibit neurite outgrowth after neuronal injury [132].

### 4.6. LncRNAs Directly Modulating Protein Function

It is more and more recognized that some lncRNAs have evolved the ability to sequester proteins and interfere with their function [133,134]. In the brain, one of the few examples of lncRNAs able to directly regulate extranuclear protein function is the lncRNA *NEAT1*, which is also found in paraspeckles with *Malat1* [123]. *NEAT1* expression is activity-dependent and its levels are high at rest while they decrease upon neuronal depolarization. Outside the nucleus, *NEAT1* binds two potassium channel-interacting proteins, KCNAB2 and KCNIP1. Downregulation of *NEAT1* combined with neuronal depolarization results in the increase of the cytoplasmic concentration of these channel-interacting proteins, thus modulating neuronal excitability. As observed in other cases, the function of lncRNAs seems to be pleiotropic, as exemplified in this case by the observation that *NEAT1* controls the expression of ion channel related genes [134] and that it can interfere with gene transcription and histone 3 lysine 9 dimethylation (H3K9me2), a histone modification that represses gene function impairing memory formation in young adult mice [135].

During the characterization of lncRNAs implicated in neurotransmitter and synaptic vesicle release, our group has identified *neuroLNC*, a nuclear lncRNA that is highly neuron-specific and that is conserved from rodents to humans [136]. *NeuroLNC* influences several aspects of neuronal physiology such as calcium influx, neurite elongation and neuronal migration in the developing brain [136]. Of note, we found that, in neurons, *neuroLNC* functions by interacting with the RNA-binding protein TAR DNA binding protein-43 (TDP-43), leading to the specific stabilization of mRNAs encoding for presynaptic proteins. Mutating the UG-repeats which are essential for the binding of *neuroLNC* to TDP-43 ablates the ability of *neuroLNC* to promote synaptic vesicle release when overexpressed. Likewise, the down-regulation of TDP-43 abolishes the effects of *neuroLNC*, reveling a clear dependency between the function of *neuroLNC* and TDP-43. Another recent example of lncRNAs associated to neuronal activity modulation is *ADEPTR*, which can be quickly produced and transported to spines in a KIF2A-dependent mechanism requiring cAMP/PKA signaling [137]. *ADEPTR* interacts with spectrin 1 and ankyrin B to regulate their localization and tune spine dynamics [137].

In neurons, the tuning of local translation is associated with activity-dependent synaptic plasticity to regulate the expression of mRNAs encoding synaptic proteins [138]. Deregulation of neuronal protein synthesis increases neuronal hyperexcitability and seizure susceptibility [139,140]. Cytoplasmic lncRNA-like molecules might be crucial players as they display roles in the regulation of local mRNA translation. As an example, BC1 or its primate analog, BC200, two neuron-specific non-coding RNAs, have been proposed to locally regulate protein translation in dendrites. *BC1* measures 152 nucleotides and *BC200* 200 nucleotides, and although often referred to as lncRNAs, since lncRNAs are defined as molecules > 200 nucleotides, we refer to these as ‘non-protein coding RNAs’. These molecules were initially described as taking part in a dendritic ribonucleoprotein (RNP) complex in, regulating translation initiation in response to synaptic stimuli by interacting with SRP9/14, eIF4A, eIF4B, PABP, FMRP and SYNCRIP that interfere with 48S initiation complex formation [141,142,143,144,145,146,147,148]. Moreover, it was shown that *BC1/BC200* RNA can inhibit translation of dendritic protein synthesis affecting ARC and CaMKII levels [145]. It is important to underline that the relevance of these results needs to be very carefully weighted, since the interaction of FMRP with *BC1/BC200* has been clearly confuted by several subsequent works, indicating that FMRP and *BC1/BC200* act independently and do not in fact bind each other specifically, neither in vitro nor in vivo [149,150,151]. Similarly, the interaction between *BC200* and *SYNCRIP* has also been argued [152]. In any case, *BC1/BC200* seems to be implicated in neuronal function since the knockout mice show hyperexcitability and neuronal plasticity changes which can probably be reconducted to altered translation of group I metabotropic glutamate receptors (mGluRs) [153].

In all these cases showcasing examples of direct interaction between lncRNAs and proteins, only few direct examples of extranuclear interactions of lncRNAs and proteins are found. It is tantalizing to speculate that some lncRNAs might be implicated in the direct regulation of phase separation and condensate formation at the synapse [154], in a similar manner to what has been observed in the nucleus [155], although direct proof is still missing. The most relevant lncRNAs acting in the brain and in neurons are summarized in Table 1.

## 5. LncRNAs in Brain Disease and Neurodegeneration: Biomarkers or Active Players?

Considering the importance of lncRNAs in neuronal physiology it is not surprising that dysregulation of lncRNAs in the brain might be linked to neurodevelopmental disorders [160,161], autism spectrum disorders [162,163,164], substance use [165], intellectual disability [166], neurodegenerative diseases and neurological disorders [167,168]. A growing number of studies report that lncRNAs are associated with the pathology and development of neurodegenerative diseases [80,114,169,170]. Since the therapeutic windows for neurodegenerative diseases are narrow and current treatments only delay their progression, understanding the etiology of these diseases would provide new therapeutic approaches. Importantly, lncRNAs can be detected in easily-accessible biological fluids including blood and plasma by sequencing technologies that are sensitive and compatible with the clinical environment, hence they seem to be reliable biomarkers as also observed in the field of cancer biology [171].

Biomarker signatures based on lncRNAs could be used for detecting neurodegenerative processes very early, when clinical symptoms are not yet apparent, and degenerative processes are not yet irreversible. Moreover, due to the difficult accessibility of brain tissue in patients, monitoring the expression of biomarker lncRNAs from peripheral fluids would allow to more reliably follow the progression of the pathology and possibly even understand which cell types are more actively contributing to the neurodegenerative process. Overall, for these reasons, lncRNAs hold great potential as biomarkers for neurodegenerative diseases. At the same time, one aspect that still requires thorough examination, is whether the expression of lncRNAs during neurodegenerative processes just correlates with brain alterations or if in some cases it might also actively contribute to the pathogenesis. In other words, if lncRNAs have a functional role in the pathology in some instances, they could be exploited as possible therapeutic targets. We will summarize here observations suggesting that only a few lncRNAs show a direct involvement in neurodegenerative alterations.

The most common neurodegenerative disease, accounting for around 80% of dementia cases in the elderly population, is Alzheimer’s Disease (AD) [172]. In the APP/PS1 mice, which are used as AD model, several lncRNAS are differentially expressed, providing a number of possible biomarkers to screen in the human pathology [173]. Inhibition of the lncRNA *BACE1-AS* in vivo drives the reduction of β-amyloid synthesis and aggregation in the brain by decreasing the level of *Bace1*. As previously mentioned, *BACE1-AS* stabilizes BACE1 which is the enzyme cleaving amyloid precursor protein (APP) and producing the amyloid β precursors. It seems that *BACE1-AS* is involved in a positive feedback loop with β-amyloid peptides, so that high levels of *BACE1-AS* in AD results in amyloid beta-peptide synthesis and, subsequently, amyloid beta-peptide induces elevated *BACE1-AS* expression [80]. *BC200*, another non-protein coding RNA, negatively regulated over aging, is significantly upregulated in AD [174]. The increased level of *BC200* and its delocalization in perikaryal structures was also correlated with AD severity [174]. The dysregulation of lncRNAs, such as *lncRNA-17A*, *51A*, and *Sox2OT* in AD are also reported in other studies [175,176,177]. Besides *BACE1-AS,* in all other examples, the correlations between lncRNA levels and the presence of pathological alterations cannot be considered sufficient indication for a direct involvement in AD and targeted studies are required to clarify if the modulation of these lncRNAs has a direct role in the modulation of the pathology.

The lncRNA *AS-Uchl1* is an antisense RNA that seems to increase the translation of UCHL1, a deubiquitinating enzyme whose reduction is associated with neurodegenerative diseases and mutations in the *Uchl1* gene are found in familial Parkinson’s disease (PD) [178,179,180,181]. In a rotenone-induced PD mouse model, the lncRNA *AS-Uchl1* was significantly downregulated in dopaminergic neurons indicating a possible direct role in the pathology [180]. Several other lncRNAs show significant changes in PD. As an example, the expression of *lincRNA-p21*, *Malat1*, *SNHG1*, *HOTAIR* and *TncRNA* are upregulated in PD, while the two lncRNAs upstream of the lncRNA *H19* (*Huc 1* and *2*) are downregulated [182,183]. *Malat1* in this context has been proposed to have a detrimental role and promote inflammasome activation and reactive oxygen species (ROS) production by interfering with the expression of NRF2, a transcription factor that controls the expression of antioxidant proteins during inflammation [184]. A recent work indicates that *HOTAIR* might act as a regulator of *miR-126-5p* which in turn targets the RAB3A interacting protein RAB3IP in PD cells and in a PD mouse model where the pathology is induced with the neurotoxin 1-methyl-4-phenyl-1,2,3,6-tetrahydropyridine (MPTP) [185]. RAB3IP in PD cells seems to worsen the pathology via a mechanism that inhibits neuron autophagy and increases apoptosis [185]. Among the lncRNAs increased in PD, *LINC-PINT* was recently found to correlate with neuronal maturation and increase in levels in several brain regions of neurodegenerative diseases [158]. Its depletion in SH-SY5Y cells leads to increased cell death, suggesting that it might have a protective role and thus is being induced due to increased cellular stress during neurodegeneration [158].

Huntington disease (HD) is an inherited neurodegenerative disorder with progressive chorea and cognitive impairment. Patient postmortem HD brains exhibited dysregulated lncRNA expression such as *NEAT1* upregulation which is potentially induced by ubiquitin-proteasome system impairment in HD [186]. *NEAT1* upregulation under stress has implications in cell survival pathways and might represent a neuroprotective mechanism against neuronal damage in HD, as also observed in relation to its ability to meliorate the proteostasis stress caused by TDP-43 [187]. In contrast, *Tuna* expression is potentially negatively associated with the severity of HD as zebrafish treated with *Tuna* antisense showed severe locomotor defects [114]. Of note, the transcription factor REST, a pivotal repressor of key target genes in HD, negatively regulates some of the differentially expressed lncRNAs in HD such as *HAR1F* and *HAR1R*, which are decreased in the striatum of HD patients [188]. Furthermore, the genes for the lncRNA *DGCR5* and *MEG3* have REST binding sites and are downregulated in HD [189].

LncRNAs have been implicated in neuronal degeneration during amyotrophic lateral sclerosis (ALS) and in motoneuron pathology [190,191]. Studying a cell line expressing a mutant FUS devoid of the nuclear localization signal and ALS-patient derived fibroblasts revealed that mutant FUS results in accumulation of *NEAT1* isoforms and excessive paraspeckle formation which may contribute to ALS severity [192,193]. Additional studies have also demonstrated the potential role of the lncRNA *ATXN2-AS* in spinocerebellar ataxia type 2 which is associated with an increased risk for ALS [194]. Screening of peripheral blood mononuclear cells from sporadic ALS patients revealed that there are 293 differentially expressed lncRNAs in patients in comparison with healthy individuals reflecting the involvement of *lncRNAs* in ALS [195]. In all these examples there is only a correlation between lncRNA expression and pathology.

Dysregulation of lncRNAs has been associated with synapse function and neurological disorders such as epilepsy. *Evf2*, as described above, is a crucial lncRNA in GABAergic synaptic transmission of medial prefrontal cortex and hippocampal neurons. In vivo analyses of *Evf2* showed that its downregulation increases seizure susceptibility and severity in the adult mouse brain [74]. In contrast, *NEAT1*, which regulates potassium channel function, is upregulated in human epilepsy samples and increases seizure susceptibility [134]. Ion channel dysregulation is also implicated in neuropathic pain. Reduced potassium current after peripheral nerve injury contributes to the neuropathic pain via *KCNA2-AS* induction, as also discussed earlier. Notably, hypersensitivity and neuropathic pain could be decreased by blocking *KCNA2-AS* expression, opening a new therapeutic window in relieving neuropathic pain [84]. The imprinted lncRNA *H19* has also been implicated in epilepsy, since its expression is strongly upregulated in a rat model of temporal lobe epilepsy [196]. Its function is probably mediated by a sponging effect on the microRNA let-7b which is a known regulator of the cell cycle in the developing cerebral cortex [196,197].

The lncRNA *lnc-NR2F1* is a conserved lncRNA that was recently found to be associated with autism spectrum disorder and intellectual disability in children and it has been observed that the *lnc-NR2F1* locus is frequently mutated in these patients [159]. Importantly, functional characterization of *lnc-NR2F1* revealed that this lncRNA binds to a genomic region enriched with basic helix-loop-helix motifs to regulate the transcription of genes involved in neuron maturation [159].

Emerging evidence suggests that lncRNAs are involved in various other psychiatric disorders including schizophrenia. Schizophrenia is a complex psychiatric disease with aberrant splicing of genes associated with this pathology, including *Disc1* and *Errb4* [198,199]. It has been shown that transcripts of these genes and their splicing variants are upregulated by *Gomafu* knockdown through an interaction with the splicing proteins SF1, SRSF1 and QKI [50,157]. *Gomafu* is also downregulated in post-mortem cortex of patients with schizophrenia [50]. In addition, the gene encoding *Gomafu* is embedded in a schizophrenia linked chromosomal region [200,201] and its genetic variant is associated with the risk of paranoid schizophrenia in a Chinese Han population [202], which altogether suggest a potential link of *Gomafu* dysregulation with schizophrenia pathogenesis.

## 6. Conclusions and Perspectives

The study of lncRNAs in neurological disorders using animal models and in vitro strategies is promising for defining the etiology of neuropathologies, and in the future will eventually contribute to the diagnosis and the treatment of brain diseases [203]. A recent study also indicates that the lncRNA *XIST* can be used to trigger silencing of chromosome 21 in Down syndrome patient-derived cells, and correct a differentiation delay observed in this model [204], opening new avenues for leveraging lncRNA biology in alleviating dosage problems in the brain. Modeling some aspects of lncRNA biology in human iPSC-derived neurons and brain organoids might be a way to overcome implementation limitations, although in vivo solutions, such as chimeric mice studies [205], will also be necessary to correctly model lncRNAs. The comparative analysis of lncRNAs across different species [53] is also a powerful approach that can be used to understand lncRNA function [18], providing information about lncRNA and genome interaction [206], and broadening the computational toolbox for the study of this molecular species. At the same time, lncRNAs are often not conserved between rodents and humans [18], rendering this task particularly complex.

One important aspect is to characterize lncRNAs in a broad and unbiased manner. Along these lines, omics studies combining lncRNA knockdown or knockout with molecular phenotyping (such as deep-sequencing libraries) are becoming available for human cell lines [207], and, although technically challenging, the application of similar approaches to iPSC-derived human neurons seems a likely future development. What renders lncRNAs particularly difficult to model and study in neurons is the fact that they often act as fine modulators of sophisticated regulatory networks and the phenotypes are subtle. For this reason, whenever possible, the best approach to study lncRNAs would be to use an inducible gene knockout model, allowing to trigger lncRNA inactivation in the right time frame thus avoiding developmental adjustment or ‘vicariation’ effects that would mask the exact role of these molecules.

In the brain, a notable recent example of the fact that lncRNAs can be part of subtle regulation of neuronal function is *Cyrano*, which promotes the degradation of *miR-7* and thus favors the accumulation of *Cdr1as*, a circular RNA that modulates neuronal activity [47]. In the absence of *Cyrano*, *Cdr1as* is depleted in neurons, in concomitance with the help of a second miRNA, *miR-671* [47]. This is just one example, and more generally the study of lncRNAs will require the concomitant measure of several omics layers to truly narrow down their most relevant molecular functions [136]. Of note, lncRNAs in the brain have been linked to memory formation and behavior, indicating that their role is important at the whole organism level, and not only for the fine tuning of neuronal activity [135,208]. Moreover, due to the limited number of functional studies, compared to other molecular species, lncRNAs lack extensive resources for their interpretation [209,210,211], although efforts are being made and these tools will become more accessible in the future.

One aspect that very often remains obscure in lncRNA biology is whether these molecules only show correlations with a specific pathology or phenotype or whether they are active players. This can only be addressed if their ablation or overexpression can recapitulate some of the observed changes that do correlate with their expression levels. In order to exclude that their effect is overestimated, direct KO in mice or cells with CRISPR approaches is necessary, although in some cases it leads to puzzling results, as in the case of *Malat1* KO mice which show no obvious differences [122,123,124]. In light of the complex interactions between lncRNAs with the genome and their roles in nuclear organization, it is necessary to apply and further develop methods for capturing these transient interactions in the most reliable manner, using crosslinking conditions that avoid artifacts [212], and more generally to adapt and further validate the techniques to reliably study these elusive biomolecules.

For the examination of neurodegenerative diseases in clinical settings, lncRNAs hold great potential since brain lncRNAs can be detected in peripheral fluids [213] and can thus be explored as biomarkers for both following disease progression and evaluating the outcome of therapeutic regimes. It is also possible that the study of the regenerative potential of the peripheral nervous system, where several lncRNAs are expressed during the regenerative period that follows injury [214], will provide new targets that could be leveraged for driving the repair of the damaged central nervous system or possibly to revert some of the changes observed in aging [215].

Overall, we foresee that a more systematic study of lncRNAs in the pathophysiology of the brain will provide a more comprehensive representation of the complexity of brain function.

## Figures and Tables

**Figure 1 molecules-26-03951-f001:**
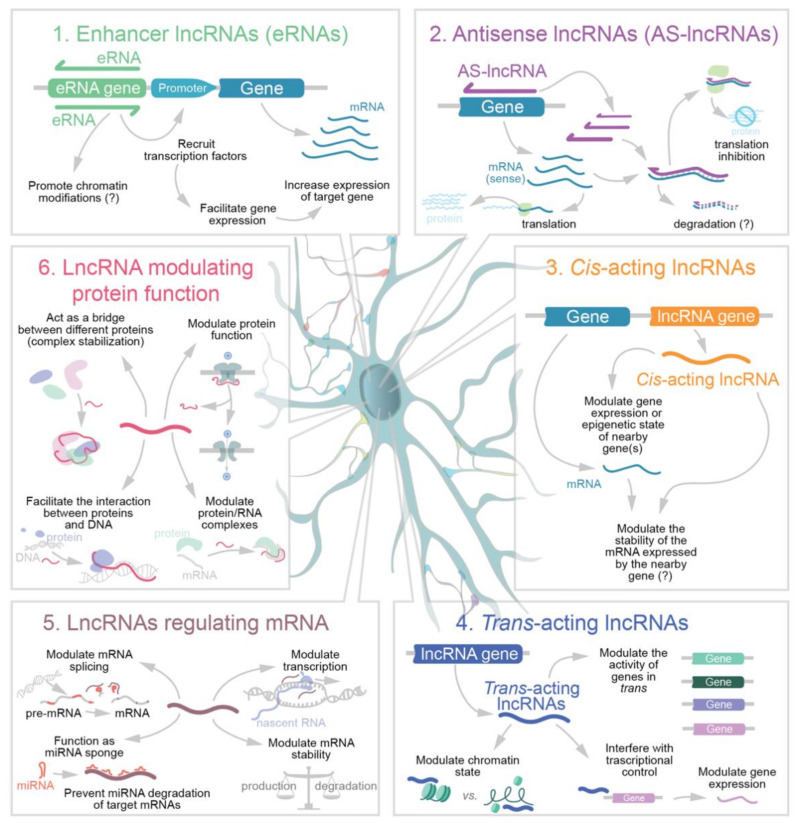
Schematic representation of the functional classes of neuronal lncRNAs.

**Table 1 molecules-26-03951-t001:** Summary of relevant neuronal lncRNAs including interactors, molecular targets and function.

Name	Biological Function	Interactor(s)	Molecular Target(s)	Cellular/Molecular effect	Reference
*ADEPTR*	Activity-dependent control of spine dynamics	Spectrin 1 and ankyrin B	Spectrin 1 and ankyrin B	Regulates the localization of structural components of spines	[137]
*Arrl1*	Inhibits neurite outgrowth	*miR-761* (*Arrl1* is a decoy of this miRNA)	Cdkn2b (inhibitor in neurite outgrowth)	Arrl1 sponges *miR-761* which would degrade *Cdkn2b*	[132]
*BC1/BC200*	Regulation of neuron plasticity and excitability	SRP9/14, eIF4A, PABP, FMRP and SYNCRIP	Dendritic transcripts (e.g., *Arc*, *CaMKII*, *MAP1B* and *mGluRs*)	Translational regulation, see text for details	[143,144,145,146,148,153]
*BDNF-AS*	Modulation of neuronal differentiation and survival	EZH2	BDNF	Chromatin remodeling and posttranscriptional regulation (NAT)	[63,90]
*Dali*	Differentiation of neuro- blastoma cells	Pou3f3, DNMT1, BRG1 and SIN3A	Neuronal differentiation genes (e.g., *Pou3f3*, *E2f2*, *Fam5b*, *Sparc*, *Kif2c*)	Chromatin remodeling and transcriptional regulation	[64]
*EVF2*	GABAergic interneuron development	DLX, MECP2, Smarca4, BAF170 and SMC1	Interneuron subtype-genes (e.g., *DLX5*, *GAD1*, *Rbm28*, *Akr1b8*, *Umad1*, *Lsm8*, *Npy*, *Sst*, *5Htr3a*)	Chromatin remodeling, scaffold, transcript regulation	[74,75,156]
*GM12371*	Regulation of neuronal differentiation	N/A	Genes involved in neuronal development and differentiation (e.g., *Sox10*, *PRKCq*)	Transcriptional regulation	[115]
*Gomafu*	Control of progenitor differentiation and survival	SF1, SRSF1 and QKI	Neuronal differentiation genes (e.g., DISC1, ERRB4, WNT7B)	Posttranscriptional regulation (alternative splicing)	[50,157]
*HOTAIRM1*	Control of neurogenesis and neuronal differentiation	N/A	Neurogenin 2	Epigenetic regulation of Neurogenin 2 locus	[113]
*KCNA2-AS*	Regulation of excitability and neuropathic pain	N/A	*KCNA2*	Posttranscriptional regulation (NAT)	[84]
*linc-Brn1b*	Regulation of neocortex development	N/A	*Brn1*	Transcriptional regulation	[11]
*LINC-PINT*	Neuroprotection in neurodegenerative diseases (?)	N/A	*Prc2* and its targets	N/A	[158]
*lnc-NR2F1*	Neuronal maturation	N/A	Various genes involved in the development of the central nervous system	Transcriptional regulation	[159]
*lncND*	Regulation of neuronal differentiation	*miR-143-3p*	*NOTCH-1*, *NOTCH-2*	miRNA sponging	[129]
*lncOL1*	Induction of oligodendrocyte maturation	Suz12	Oligodendrocyte precursor cells gene program	Chromatin remodeling	[112]
*lncRNA-1604*	Regulation of neural differentiation	N/A	ZEB1/2, *miR-200C*	Post-transcriptional regulation (miRNA sponging)	[130]
*Malat1*	Regulation of synaptogenesis and neuritogenesis	SF2/ASF-SC35(co-localization)	Genes involved in synapse and dendrite development	Transcriptional and alternative splicing regulation	[120,121]
*NEAT1*	Regulation of neuronal excitability	KCNAB2, KCNIP1, TDP-43 and FUS/TLS	Genes involved in ion channel activity	Transcriptional regulation, protein localization	[134]
*NeuroLNC*	Regulation of neuronal differentiation and activity	TDP43	mRNAs for synaptic proteins	Transcriptional regulation (?)	[136]
*Paupar*	Differentiation of neuro- blastoma cells, neurogenesis	Pax6, KAP1	Genes involved in stem cell self-renewal and different. (e.g., *Pax6*, *SOX2*, *HES1 and EVF2*)	Chromatin remodeling and transcriptional regulation	[104,106]
*PNKY*	Regulation of NSC self-renewal and differentiation	PTBP1	Genes involved in cell adhesion, synaptogenesis and neurogenesis	Transcriptional modulation and alternative splicing regulation	[127]
*RMST*	Regulation of neuronal differentiation	SOX2, hnRNPA2/B1	Neurogenic TFs (e.g., *DLX1*, *ASCL1*, *HEY2* and *SP8*)	Transcriptional regulation	[117]
*SIX3OS*	Retinal and neuronal cell lineage specification	EZH2 and EYA	SIX3 target genes	Chromatin remodeling and transcriptional regulation	[41,98]
*Synage*	Cerebellar. synapse development	*miR-325-3p*	Cerebellin 1 Precursor (*Cbln1*)	miRNA sponge and cellular scaffold	[131]
*Tuna*	Regulation of pluripotency and neuronal differentiation	hnRNPK, PTBP1 and NCL	Pluripotency and neuronal differentiation genes (e.g., *Nanog*, *Sox2*, *Fgf4*)	Chromatin remodeling and transcriptional regulation	[114]
*UtNgn1*	Regulation of neuronal differentiation	N/A	*Neurog1*	Transcriptional regulation	[73]

## Data Availability

Not applicable.

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
