# Peer review of "Brain Long Noncoding RNAs: Multitask Regulators of Neuronal Differentiation and Function"

_molecules, 2021, doi:10.3390/molecules26133951_

Round 1

Reviewer 1 Report

Long non-coding RNAs (lncRNAs) are ncRNAs composed of >200 nucleotides and have many functions due to their ability to bind to both proteins and nucleic acids. lncRNAs play important roles in brain development, neuron function and maintenance and are considered as modulators of many cellular functions

Several papers on the role of lncRNAs in the brain have recently been published. In this context, a review that summarizes the data is certainly useful.

In the review “Brain long noncoding RNAs: multitask regulators of neuronal differentiation and function” the authors show that lncRNAs play key roles in several regulatory mechanism in the brain. The text is complete, well-written and the bibliography is satisfactory.

Author Response

We thank the reviewer for the overall positive evaluation of our work.

Reviewer 2 Report

This is another review about the many assumed and few established functions of RNAs in the nervous system. What would be really useful to readers are critically reviews, although I concede that a critical assessment of all findings in a review would take at least ten times more work. Therefore, such reviews are far and in between, because it is much easier to list the suggested functions instead of painstakingly follow the subsequent citations in order to determine, whether a proposed function has stood the test of time or whether there are controversial findings. Also, it should be stated that most of the time a gene knockout is to be preferred over a gene knockdown.

On line 238 or 467, there are examples on how it should be done. Such critical assessment is absent in most other cases, such as for the prominent controversy about FMRP (and also SYNCRIP) binding to the BC RNAs (lines 454-458). This review fails to mention several studies from other laboratories that could not confirm binding of FMRP to BC1 or BC200 RNA (PMID: 18184799 ). Darnell et al. (PMID: 15805463) already found the following: “We tested whether an excess of BC1 RNA or tRNA was able to compete FMRP off polyribosomes (Fig. 8). Addition of up to 5 μM BC1 or tRNA to cortical lysates fails to compete with the in vivo binding site of FMRP on polyribosomes and displace it (Fig. 8B,C). These results demonstrate that FMRP is specifically associated with brain polyribosomes, and that this interaction is effectively competed by kissing complex RNAs.” Yan and Denman (PMID: 21772992) concluded:  “Given this, our data generated using different constructs, different preparations and different methods, converge with the published work of Iacoangeli et al. [47] to support a model in which FMRP and BC1 RNA operate independently of each other to control protein synthesis in neuronal processes.” And, finally, Booy et al. (PMID: 30247708) state: “Amongst the eight excluded proteins was the previously reported binding partner, FMR1. Furthermore, SYNCRIP bound to only a small fraction of the input BC200 and demonstrated marginal enrichment.” If problems like this are not being addressed in reviews, undead artifacts can haunt a scientific field for an unnecessarily long time. As an aside, BC1 RNA and BC200 RNA are not homologs but functional analogs (line 454). Homology implies a common evolutionary origin. While rodent BC1 RNA originated from a tRNA, BC200 RNA originated from SRP RNA via a monomeric Alu element (PMID: 25081515).

The BC RNAs highlight a minor, more formal problem. BC1 RNA, even BC200 should not be mentioned here because it is not a long RNA but a short RNA by the unfortunate newer definition of the 200nt and below cutoff. Traditionally, even longer RNAs (SRP RNA, 7SK RNA of  >300 nt) were considered small RNAs. And while I am at it, the term noncoding RNA should rather be nonprotein coding RNA as most RNAs carry some sort of code (PMID: 25081515, PMID 26818079), but this is a lost cause.

Also, the statement that “more than ~80% [of the genome] is involved in the production of a variety of noncoding RNAs” cannot go unchallenged. Apart from the fact that the act of transcription alone can regulate the expression of neighboring genes (PMID: 29309647, PMID: 32133533) (lines 84, 97) with the by-product of non-functional transcripts (that, however, could be raw-material for the evolution of de novo genes), much of the genomic transcription is probably noise (PMID: 23431001, PMID: 31697388, PMID: 15851065, PMID: 26818079).

Line 456 – 457, inhibiting translation initiation in response to synaptic stimuli: This occurs via eIF4B, which is not mentioned (Eom et al., PMID: 25332164). Importantly, after synaptic stimulation there is disinhibition, and not inhibition (see once more Eom et al.).

A problem in the manuscript is the lack of consistency with respect to italics and non-italics. In gene designations, the italic form should be used. When you write about a gene product – whether protein or RNA – the non-italic form should be used throughout the manuscript.

Other queries:

Line 34: “actively transported” is there also passive transport?

Line 36: instead of sophisticated tissues, perhaps complex tissue (all tissues are sophisticated)

Line 44: small nuclear RNAs are not involved in translation (pre-mRNA processing)

Lines 53-54: “initially devoid (of open reading frames) are actually involved in the production of small peptides” – something missing, like the words in parentheses

Line 84” synaptic plasticity?

Line 87: ref. 43 is a comparison human/macaque and not mouse and hence not surprisingly sequence conserved; similar line 94/95

Line 367: synaptic transcripts (there is – to the best of my knowledge – no transcription at synapses; perhaps transcripts transported to synapses?

Line 415: what is “highly cytosolic”?

Line 524: what is “the two conserved elements upstream of the lncRNA H19”? What elements?

Line 552: what is a FUS cell line?

Line 606: better: …not conserved between rodents and humans

Author Response

We have uploaded a .pdf file containing the point-by-point response to the Reviewer’s comments.

Reviewer 3 Report

The review extensively summarizes the current knowledge of the ncRNA types, structure and functions and put that into the context of the nervous system development – both physiological and pathological. The article is well and logically organized and presents comprehensibly the fundamental information about ncRNA. The scientific relevance of the review is high and reflect the current interest into the ncRNA metabolism. With regards to the quality of the review I do recommend it for the publication in the current form.

Author Response

(The authors gave the same response as above.)

Round 2

Reviewer 2 Report

Thanks to the authors for their detailed response. I just caught one typo around line 644/645:

"...inactivation in the right time frame..."